

# Water stress and nitrogen supply affect floral traits and pollination of the white mustard, *Sinapis alba* (Brassicaceae)

Asma Akter[1,2] and Jan Klečka[2]

[1] Department of Zoology, Faculty of Science, University of South Bohemia, České Budějovice, Czech Republic
[2] Institute of Entomology, Biology Centre of the Czech Academy of Sciences, České Budějovice, Czech Republic

## ABSTRACT

Changes in environmental conditions are likely to have a complex effect on the growth of plants, their phenology, plant-pollinator interactions, and reproductive success. The current world is facing an ongoing climate change along with other human-induced environmental changes. Most research has focused on the impact of increasing temperature as a major driving force for climate change, but other factors may have important impacts on plant traits and pollination too and these effects may vary from season to season. In addition, it is likely that the effects of multiple environmental factors, such as increasing temperature, water availability, and nitrogen enrichment are not independent. Therefore, we tested the impact of two key factors—water, and nitrogen supply—on plant traits, pollination, and seed production in *Sinapis alba* (Brassicaceae) in three seasons defined as three temperature conditions with two levels of water and nitrogen supply in a factorial design. We collected data on multiple vegetative and floral traits and assessed the response of pollinators in the field. Additionally, we evaluated the effect of growing conditions on seed set in plants exposed to pollinators and in hand-pollinated plants. Our results show that water stress impaired vegetative growth, decreased flower production, and reduced visitation by pollinators and seed set, while high amount of nitrogen increased nectar production under low water availability in plants grown in the spring. Temperature modulated the effect of water and nitrogen availability on vegetative and floral traits and strongly affected flowering phenology and flower production. We demonstrated that changes in water and nitrogen availability alter plant vegetative and floral traits, which impacts flower visitation and consequently plant reproduction. We conclude that ongoing environmental changes such as increasing temperature, altered precipitation regimes and nitrogen enrichment may thus affect plant-pollinator interactions with negative consequences for the reproduction of wild plants and insect-pollinated crops.

Corresponding author
Asma Akter, asma.akter@entu.cas.cz, asma.akter84@gmail.com

## INTRODUCTION

Ecosystems worldwide are facing accelerating global change characterised by increasing temperature and changing levels of precipitations, coupled with an increasing supply of

nitrogen and other nutrients, biological invasions, and habitat loss (*Hoover et al., 2012*). Testing the effects of these environmental changes on plant growth and reproductive fitness are necessary to understand potential impacts of climate change on the productivity and functioning of natural and agricultural ecosystems (*Rustad, 2008*). For pollinator-dependent plants, changes of these factors may impact their relationship with pollinators and consequently the success of pollination and plant reproduction (*Scaven & Rafferty, 2013*; *Gérard et al., 2020*). Reproductive success of animal-pollinated plants generally depends on floral traits, which act as an advertisement of rewards to their pollinators (*Hegland & Totland, 2005*; *Basnett, Ganesan & Devy, 2019*). Despite strong selection from pollinators, plant populations naturally show significant variation in their morphological, phenological, and floral traits. A part of this variation results from heritable genetic differences among individuals, while the rest (phenotypic plasticity) is caused by local environmental factors (*Holtsford & Ellstrand, 1992*; *Gray & Brady, 2016*). Changing environmental factors may thus alter plant–pollinator interactions as a consequence of changing plant traits (*Carroll, Pallardy & Galen, 2001*; *Scaven & Rafferty, 2013*; *Majetic et al., 2017*; *Rusman et al., 2019*).

Increasing temperature and water stress can have a major effect on the physiological and phenological development of plants (*Schweiger et al., 2010*). The global average annual temperature is rising gradually, with higher increases in the average and minimum temperatures reported during the winter than the summer months (*NOAA, 2021*). Consequently, phenological shifts are visible in many early-flowering plants (*Kehrberger & Holzschuh, 2019*). Increased average temperatures may allow them to initiate growth and flowering earlier because of earlier snowmelt and higher spring temperatures (*Fitter & Fitter, 2002*; *Güsewell et al., 2017*).

Water availability is changing in a complex way as many regions of the world are facing water scarcity and other regions are facing increased precipitation (*Christensen et al., 2007*). Although water availability is an important determinant of plant growth, its effect on floral traits is less clear. Water availability can directly influence the flowering time and duration (*Bernal, Estiarte & Peñuelas, 2011*; *Lasky, Uriarte & Muscarella, 2016*) and plants with adequate water supply may achieve greater height and floral abundance (*Galen, 2000*; *Carromero & Hamrick, 2005*), increased nectar production (*Zimmermann & Pyke, 1988*; *Carroll, Pallardy & Galen, 2001*), and higher nectar sucrose content (*Wyatt, Broyles & Derda, 1992*). Water stress may lead to reduced floral resources (*Rering et al., 2020*) including impaired pollen and seed development (*Barnabás, Jäger & Fehér, 2008*; *Hedhly, 2011*; *Snider & Oosterhuis, 2011*). Changes in water availability can also affect various stages of phenological growth differently in the same species (*Blum, 1996*). The consequences of water stress for flower visitation by insect pollinators are still poorly known, but there is evidence that alteration of floral rewards by water stress may lead to decreased flower visitation (*Descamps et al., 2018*).

Variation in nitrogen supply is another key driver of local plant diversity (*Bobbink, Hicks & Galloway, 2010*). At the global scale, anthropogenic nitrogen deposition increased more than 10 times over the last century (*Galloway et al., 2004*; *Fowler et al., 2013*) and is now around 200 Tg N per year with a wide range of negative environmental consequences

(*Battye, Aneja & Schlesinger, 2017*). In a plant community, small-scale heterogeneity of soil nitrogen content at the scale of a few meters can lead to variation in plant size and reproductive success (*Scott-Wendt, Chase & Hossner, 1988*), possibly including their mutualistic relationship with pollinators. Under the conditions of nitrogen limitation, increasing nitrogen supply can enhance plant growth and enable plants to produce floral rewards of higher quality (*Gardener & Gillman, 2001*; *Burkle & Irwin, 2009*; *Burkle & Irwin, 2010*). In particular, nitrogen enrichment can increase the amount of nectar produced per flower and alter the concentration and composition of amino acids in nectar, which may affect pollinator preferences and foraging behaviour (*Pyke, 1984*; *Baker & Baker, 1983*; *Inouye & Waller, 1984*; *Gardener & Gillman, 2002*; *Hoover et al., 2012*).

Changing environmental conditions can alter plant traits and disrupt interactions of plants with pollinators, but the consequences for pollination and seed production remain poorly known. Given the pace of ongoing climate change, which alters not only the temperature but also water availability for plants, and still increasing anthropogenic nitrogen deposition, it is important to investigate how these factors act interactively. Such interactive effects have been rarely considered in experimental studies on plant–pollinator interactions *Hoover et al., 2012*. To fill this gap, we examined the interactive impact of water and nitrogen supply on vegetative and floral traits, pollination, and seed production in *Sinapis alba* across three different temperature ranges coupled with seasons. It is an economically important crop, cultivated over a wide geographic range for oil and fodder and is partly self-incompatible, seed production being strongly dependent on pollination by insects. In our study we aimed to answer the following questions: (1) What are the interactive effects of water and nitrogen supply on vegetative and floral traits of *S. alba*? (2) Are these effects consistent in different temperature ranges? (3) How does intraspecific trait variation caused by growing conditions affect flower visitation by pollinators, pollination efficacy, and seed production?

## MATERIALS & METHODS

### The experimental plant, *Sinapis alba*

*Sinapis alba* (white mustard) is a rapidly growing annual plant from the Brassicaceae family with a short vegetation period. This crop is widely cultivated for seeds, oil, fodder, or as a catch crop. Flowers are yellow, produced in an elongated raceme, have four petals, four sepals, and six stamens, of which four are long and two are short. Fruit is a pod with usually four seeds but can have up to eight seeds (*Jauzein, 2011*). A wide range of pollinating insects visit this plant but the European honey bee (*Apis mellifera*), bumble bees and solitary bees are the main pollinators in Europe (*Flacher et al., 2020*).

### Growing *S. alba* under variable conditions in the greenhouse

This experiment was conducted in the same greenhouse, where *S. alba* seedlings were grown in the spring 2017 (60 plants per treatment), winter 2017–2018 (30 plants per treatment) and summer 2018 (45 plants per treatment). The number of plants grown in the first batch (spring 2017) was higher as a precaution against possible mortality of the plants and was reduced in next two batches according to the required number of plants (less

plants needed in the winter when we did no outdoor experiments). Although, the number of plants grown in the greenhouse varied in different period, but number of plants used for different data collection was not significantly different. The temperature in the greenhouse fluctuated in a near-natural way and was on average 21 °C in the winter 2017–2018, 25 °C in the spring 2017, and 29 °C in the summer 2018. A minimum 10 h of daylight (natural daylight + artificial light) was maintained for all growing conditions. Plants grown in the winter received 10-12 h of daylight, plants grown in the spring received 12-16 h, and plants grown in the summer received 16 h of daylight. The plants were grown in a combination of garden soil: compost soil: sand = 2:2:1 by volume. We analysed the total nitrogen and phosphorus content of four soil samples before the experiments, where average N content was 4.92 g/kg (SD = 1.23) and P content was 1.39 g/kg (SD = 0.12). Seeds were sown at the same time in germination trays and received the same amount of water. Seedlings were transferred to individual pots, one seedling per pot with a pot size $11 \times 11 \times 11$ cm, after four days of germination.

We divided the seedlings from the same temperature range into four treatment groups which received a different combination of two watering regimes and two levels of nitrogen supply. The average annual precipitation in the České Budějovice region in the Czech Republic is ca. 690 mm (source: Czech Hydrometeorological Institute, https://www.chmi.cz/), therefore, the corresponding water application to one pot with the size of 121 cm$^2$ would be 22.9 ml per day. So, we determined the lower level of water for one pot as 20 ml and the higher level of water as 40 ml per day, also following a trial with 10 ml, 20 ml, 30 ml and 40 ml. Deciding on a suitable level of N application was not straightforward because the ideal fertilisation depends on nitrogen availability in the soil and on precipitation (*Brown, Davis & Esser, 2005*; *Quemada & Gabriel, 2016*). The minimum N application recommended for *S. alba* is 280 kg N ha$^{-1}$ with a 560 mm annual precipitation according to *Brown, Davis & Esser (2005)* and a linear increase in seed yield was observed over the range of N application between 0 and 224 kg N ha$^{-1}$ by *DuVal (2015)* in a wetter climate of Oregon. Because of the relatively high N content of our soil mixture, we applied NPK fertiliser corresponding to 0.242 g N pot$^{-1}$ ($\sim$200 kg N ha$^{-1}$) as a higher level of application and 0.121 g N pot$^{-1}$ ($\sim$100 kg N ha$^{-1}$) as a lower level to ensure adequate N supply to the plants. We divided the total amount (*i.e.,* 0.121 or 0.242 g N) into 8 weekly doses to ensure continuous supply of N. P and K supply was the same in both levels of fertilisation (in total 0.077 g P pot$^{-1}$ and 0.033 g K pot$^{-1}$), so only the nitrogen amount differed between the two fertilisation regimes. Before conducting the main experiment, we performed a preliminary trial to determine the described water level, N application, and soil mixture to optimize the growth conditions for *S. alba*. The position of plants in the greenhouse was altered regularly to avoid any possible impact of environmental gradients, *e.g.,* the light level, within the greenhouse. A small number of plants infected with diseases or attacked by aphids were discarded from the experiment.

## Assessment of plant morphology, phenology and nectar production

We measured several vegetative and floral traits of individual plants in each treatment group following the standard BBCH scale for oilseed rape (*Meier et al., 2009*). Plant height

was measured several times throughout the growing period. Top leaves were held up together to measure the plant height until the inflorescences became taller than that and final height was taken after the end of flowering by holding up all the main and secondary inflorescence together. The number of leaves was counted for the main shoot only, as for the low water treatments there was no side shoot formation. Stem diameter of each plant was measured 20 cm above ground, up to this point, there was no side shoot formation in any treatment. The onset of flowering was counted from the day of the seedling transfer to the opening of first flower and the total number of flowers bloomed were counted until the end of flowering. Nectar was collected from four flowers per plant in each treatment group after one day of flowering by using calibrated 0.5 µl capillary tubes (Drummond Microcaps®), which allowed us to measure the volume of nectar. Additional data were collected on the aboveground fresh weight and dry weight of individual plants grown in the spring to determine the effect of different combinations of water and nitrogen availability on plant biomass. Overall, we collected data on plant height, the number of leaves, and stem diameter from 11–20 plants, sampled nectar from 15–25 plants, measured the onset of flowering in 15–30 plants, and counted the total number of flowers in 12–15 plants per treatment. All data collected during the experiments are deposited in Figshare (https://doi.org/10.6084/m9.figshare.13317686).

## Pollination efficacy treatment and field pollination observations

To determine the impact of water and nitrogen supply on the pollination efficacy in *S. alba*, we carried out self- and cross-pollination in 6 plants per treatment grown in the winter. For each treatment, 20–85 flowers were hand pollinated per plant, depending on the number of flowers produced (see data: https://doi.org/10.6084/m9.figshare.13317686). We marked all the flowers selected for the experiment, performed cross-pollination by transferring pollen from a different plant and same flower for self-pollination. We collected data on the number of fruits and seeds from each plant after three weeks to allow sufficient time for seed development. Plants grown in the spring and the summer were brought outside and placed in a sunny location nearby the Institute to assess the pollinator response and natural pollination efficacy under field conditions. First part of the pollinator observations was carried out from May 17–May 31, 2017 and a second part in from July 26–July 31, 2018. We always exposed four plants at the same time (one plant from each of the four water x nitrogen combinations), placed 1 meter apart in a square configuration. We observed their visitation by naturally occurring pollinators for 30 min, from 9:00 to 14:00 each day. Altogether, we carried out observations of forty-four groups of four plants, which resulted into a total 22 h of observation. Pollinators were observed, collected, and identified in the field. After the end of each observation, the plants were brought back to the greenhouse, their open flowers were marked, and seed production through the natural pollination was measured after seed development (data: https://doi.org/10.6084/m9.figshare.13317686).

## Statistical analyses

We used generalised linear models (GLM) to assess the individual and interactive impact of water, N, and season on the plant vegetative and floral traits. The availability of water,

nitrogen, and the season were used as factors in the analyses. Depending on the type of the response variable, we specified the GLM with either Gaussian error distribution, overdispersed Poisson ("quasipossion", the number of flowers), or Gamma distribution with a log link function (onset of flowering and nectar volume). We analysed data from the outdoor flower visitation experiment using GLM with water, nitrogen level, and season as factors, using the overdispersed Poisson ("quasipossion") error distribution. We analogously analysed also data on seed set of plants depending on growing conditions. We always examined the distribution of residuals to verify that the models fitted the data well. We conducted all analyses in R Version 3.6.3 (*R Core Team, 2020*). Most plots were created using GraphPad Prism (Version 6.01, for Windows, GraphPad Software, San Diego, California USA, http://www.graphpad.com).

# RESULTS

## Vegetative traits

We observed a complex response of the selected vegetative traits of *S. alba* to differences in the growing conditions (Table 1, Fig. 1). Plant height was affected by the three-way interaction of water availability, nitrogen supply, and season, *i.e.,* the effect of each variable was dependent on the values of the other two variables ($F = 4.94$, $P = 0.008$, Fig. 1A). While higher water availability made the plants taller, increasing nitrogen availability made the plants shorter. The magnitude of these effects varied across the three temperature ranges (Fig. 1A). In addition, plants grown in the highest temperature were almost 50% shorter than those grown in the lowest temperature. Stem diameter was larger in plants grown under high water availability ($F = 40.84$, $P < 0.001$, Fig. 1B) and varied also depending on nitrogen supply in interaction with the season ($F = 9.51$, $P < 0.001$). Higher nitrogen supplies increased stem diameter in plants grown in the lowest and highest temperature, but not in the moderate temperature (Fig. 1B). On the other hand, the number of leaves was affected only to a limited degree by water availability ($F = 16.20$, $P < 0.001$, Fig. 1C) and the temperature ($F = 7.48$, $P < 0.001$), while nitrogen supply had no measurable effect ($F = 0.09$, $P = 0.768$). Finally, dry weight of the plants grown under high water availability was 2.6 times higher compared to those grown under low water availability, with a positive effect of nitrogen supply only at high water availability (water x nitrogen interaction: $F = 5.52$, $P = 0.023$, Fig. 1D).

## Phenology and flower resource variation

The onset of flowering was significantly delayed in the lowest temperature, by ca. 21 days, compared to the moderate and highest temperature ranges ($F = 69.45$, $P < 0.001$) and also delayed by high nitrogen supply, but only by on average 3.7 days ($F = 6.21$, $P = 0.013$, Table 2, Fig. 2A). The number of flowers produced over the plants' flowering period was highly positively affected by water supply ($F = 24.24$, $P < 0.001$) and negatively by nitrogen supply ($F = 6.83$, $P = 0.010$). The effects of these two variables did not vary significantly among the time periods (Table 2, Fig. 2B). Finally, nectar volume per flower showed a complex dependence on the interaction of water, nitrogen supply, and temperature ($F = 3.56$, $P = 0.030$). Higher water availability increased nectar volume in the winter, but
**Table 1** **The effects of water availability, nitrogen supply, and season on selected vegetative traits of _S. alba_.** *F* and *P* values for individual variables and their interactions estimated by generalised linear models (see Methods) are shown.

| Variable | Plant height | | Stem diameter | | Number of leaves | | Dry weight | |
|---|---|---|---|---|---|---|---|---|
| | *F* | *P* | *F* | *P* | *F* | *P* | *F* | *P* |
| Water | **57.63** | **<0.001** | **40.84** | **<0.001** | **16.20** | **<0.001** | **105.72** | **<0.001** |
| Nitrogen | **13.44** | **<0.001** | 2.51 | 0.115 | 0.09 | 0.768 | 0.22 | 0.643 |
| Season | **62.95** | **<0.001** | **30.13** | **<0.001** | **7.48** | **<0.001** | – | – |
| Water ×Nitrogen | 9.64 | **0.002** | 0.20 | 0.660 | 0.48 | 0.491 | **5.52** | **0.023** |
| Water ×Season | 2.359 | 0.098 | 1.47 | 0.234 | 1.30 | 0.275 | – | – |
| Nitrogen ×Season | 0.457 | 0.634 | **9.51** | **<0.001** | 0.10 | 0.913 | – | – |
| Water ×Nitrogen ×Season | **4.94** | **0.008** | 0.63 | 0.532 | 0.05 | 0.951 | – | – |

**Notes.**
Bold numbers indicate significant results for *F* and *P* values.

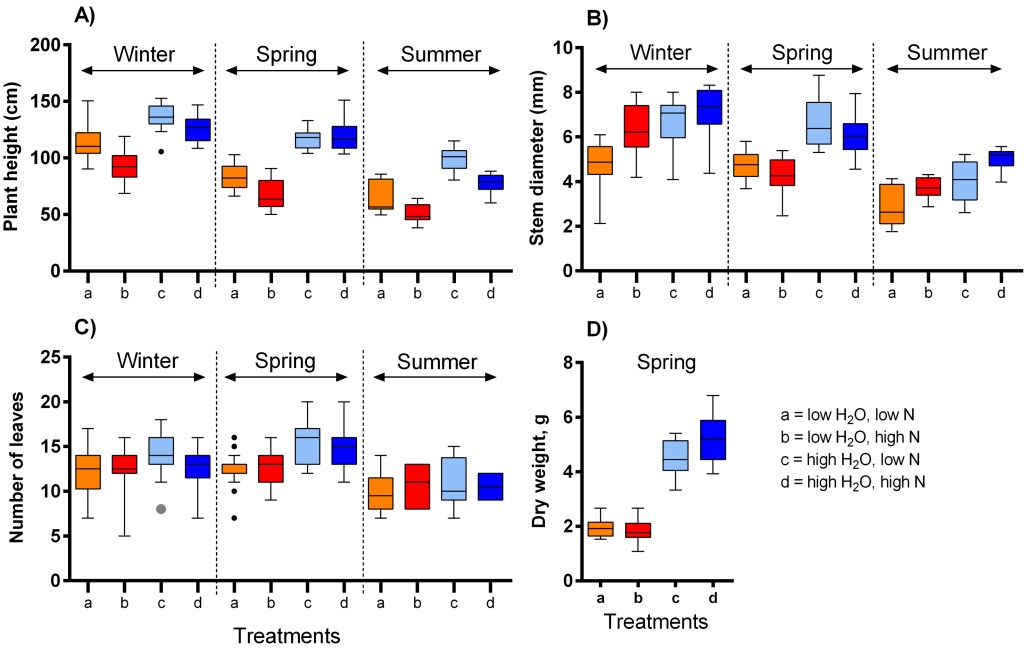

**Figure 1** **The impact of water and nitrogen supply on vegetative traits of _S. alba_ across three seasons.** Four vegetative traits were measured: (A) plant height, (B) plant diameter, (C) the number of leaves, and (D) dry weight. The boxplots show the median and interquartile range. The results of statistical tests are summarised in Table 1.

not in other times, while higher nitrogen availability increased nectar volume under low water availability in higher temperatures (Fig. 2C).

## The dependence of flower visitation on growing conditions

We observed flower visitation by eight major types of flower-visiting insects in the spring 2017 and summer 2018 which we distinguished as: the honeybee (*Apis mellifera,* in total 20 individuals), solitary bees (84), wasps (17), bumblebees (4), rapeseed beetles (*Brassicogethes*

**Table 2  The effects of water availability, nitrogen supply, and season on floral traits of *S. alba*.** *F* and *P* values for individual variables and their interactions estimated by generalised linear models (see Methods) are shown.

| Variable | Day of first flower | | Number of flowers | | Nectar volume | |
|---|---|---|---|---|---|---|
| | *F* | *P* | *F* | *P* | *F* | *P* |
| Water | 0.40 | 0.528 | **24.24** | **<0.001** | 0.06 | 0.802 |
| Nitrogen | **6.21** | **0.013** | **6.83** | **0.010** | **53.86** | **<0.001** |
| Season | **69.45** | **<0.001** | **16.12** | **<0.001** | **10.74** | **<0.001** |
| Water × Nitrogen | 0.07 | 0.790 | 3.60 | 0.060 | **21.95** | **<0.001** |
| Water × Season | 0.23 | 0.792 | 1.92 | 0.150 | 1.15 | 0.318 |
| Nitrogen × Season | 0.90 | 0.401 | 0.85 | 0.431 | **8.75** | **<0.001** |
| Water × Nitrogen × Season | 0.052 | 0.950 | 1.25 | 0.288 | **3.56** | **0.030** |

**Notes.**
Bold numbers indicate significant results for *F* and *P* values.

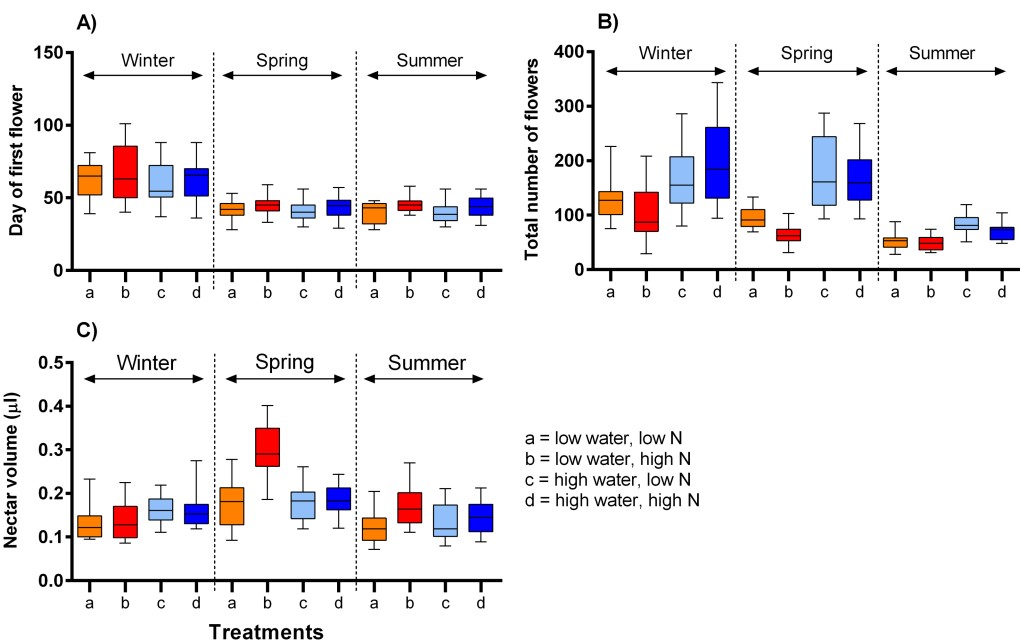

**Figure 2  The impact of water and nitrogen supply on floral traits of *S. alba* across three seasons.** (A) The onset of flowering (the day of the opening of the first flower), (B) the number of flowers produced per plant, and (C) nectar production (median and interquartile range is shown). The results of statistical tests are summarised in Table 2.

(=*Meligethes*) sp., 58), other beetles (5), hoverflies (22), and other flies (18). Rapeseed beetles were the most abundant flower visitors in the spring 2017, followed by honeybees, while solitary bees were dominant in the summer 2018, followed by hoverflies (data: https://doi.org/10.6084/m9.figshare.13317686).

Plants grown with high amount of water were visited more frequently than the plants grown with low amount of water ($F = 23.57$, $P < 0.001$) and the total number of flower visitors was higher in the spring than in the summer ($F = 14.19$, $P < 0.001$) (Figs. 3A, 3B).

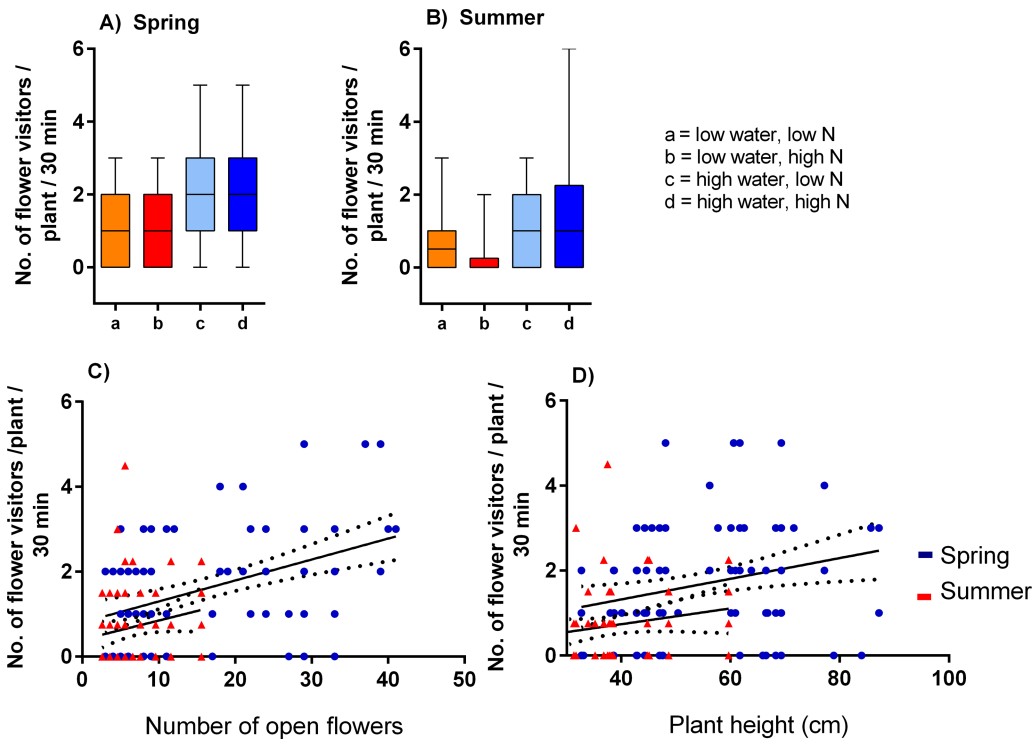

**Figure 3  Flower visitation of plants grown under varying water and nitrogen availability.** The number of flower visitors per plant per 30 min during two observation periods are shown: (A) spring 2017 and (B) summer 2018 (median and interquartile range is shown). Flower visitation also varied depending on the number of open flowers (C) and plant height (D).

Nitrogen supply under which the plants were grown did not consistently affect their flower visitation ($F = 0.26$, $P = 0.612$). Flower visitation was also affected by the number of open flowers ($F = 18.92$, $P < 0.001$, Fig. 3C) and by plant height ($F = 8.89$, $P = 0.003$, Fig. 3D), but the effect of water availability and season remained significant even after accounting for the variation in flower number and height (GLM, $F = 5.46$, $P = 0.021$ for the effect of water and $F = 6.58$, $P = 0.011$ for the effect of the season), *i.e.*, the differences in flower visitation between plants grown under different conditions could not be explained simply by differences in plant height and flower number. In addition to differences in total flower visitation, we detected changes in the composition of the flower visitors observed on plants grown under different water availability according to a redundancy analysis (RDA) performed separately for observations from the spring ($F = 4.0$, $P = 0.004$) and summer ($F = 3.1$, $P = 0.028$), while nitrogen supply did not affect the composition of flower visitors ($F = 0.37$, $P = 0.869$ for the spring data and $F = 0.57$, $P = 0.669$ for the summer data). Some flower visitors visited plants grown under high water availability more frequently, particularly solitary bees and rapeseed beetles, while other flower visitors did not show a clear preference (Figs. 4 and 5).

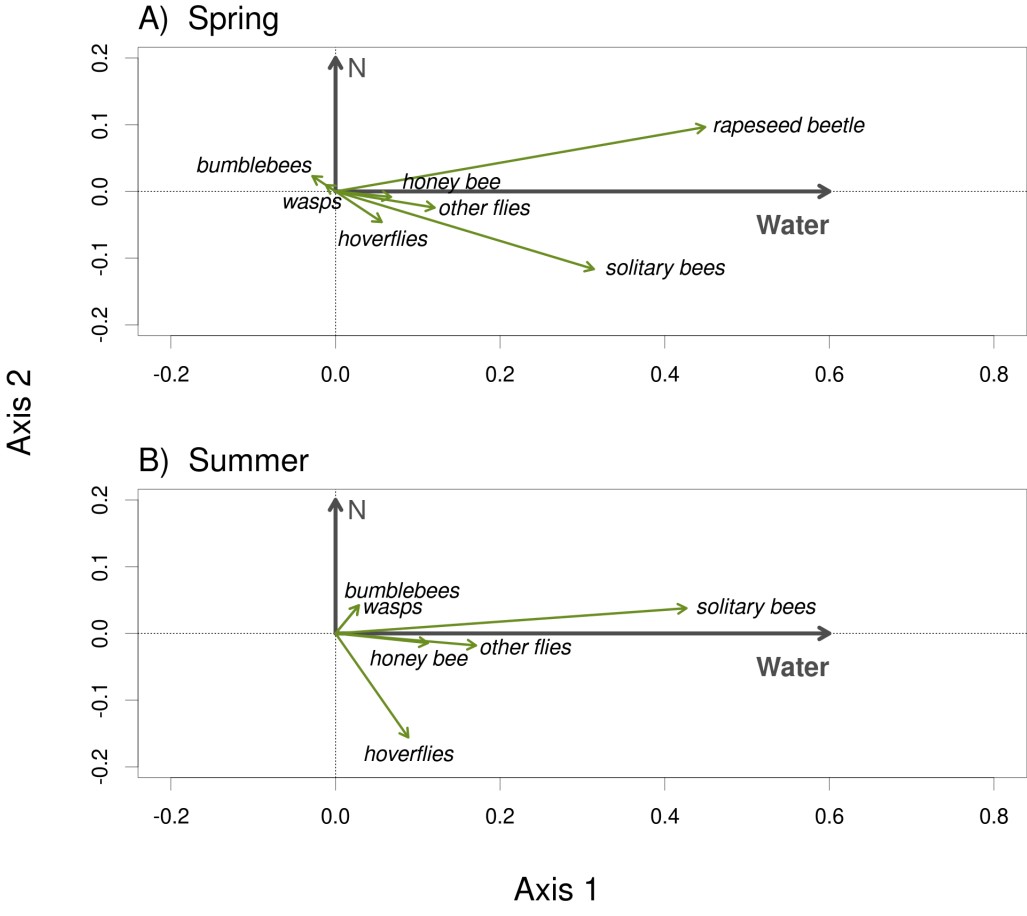

**Figure 4 The composition of the flower visitor community varied depending on water availability under which the plants were grown.** The results of RDA show the effect of water availability and the lack of an effect of nitrogen supply during the plant growth period on the composition of the flower visitor community in the spring 2017 (A) and the summer 2018 (B).

## Pollination efficacy and seed production

Our hand pollination experiment confirmed that *S. alba* is partially self-incompatible. Plants cross-pollinated by hand using a brush produced ca. 3.9 times higher number of seeds per flower than the self-pollinated ones; on average 2.9 compared to 0.7 seeds per flower (Fig. 6A). However, the seed set depended not only on the mode of pollination (self-pollinated compared to cross-pollinated) but on its interaction with nitrogen availability ($F = 10.64$, $P = 0.002$). Specifically, higher nitrogen availability increased seed set in self-pollinated plants, but decreased seed set in cross-pollinated plants. In addition, higher water availability increased seed set in both self-pollinated and cross-pollinated plants irrespective of the nitrogen level ($F = 5.24$, $P = 0.028$) (Fig. 6B).

Plants exposed to natural pollination in the spring and summer produced a variable number of seeds per flower depending on the interaction of water availability and season ($F = 14.74$, $P = 0.0003$). We observed a slightly higher seed production per flower in plants grown under high water availability in the summer 2018, but no significant difference in

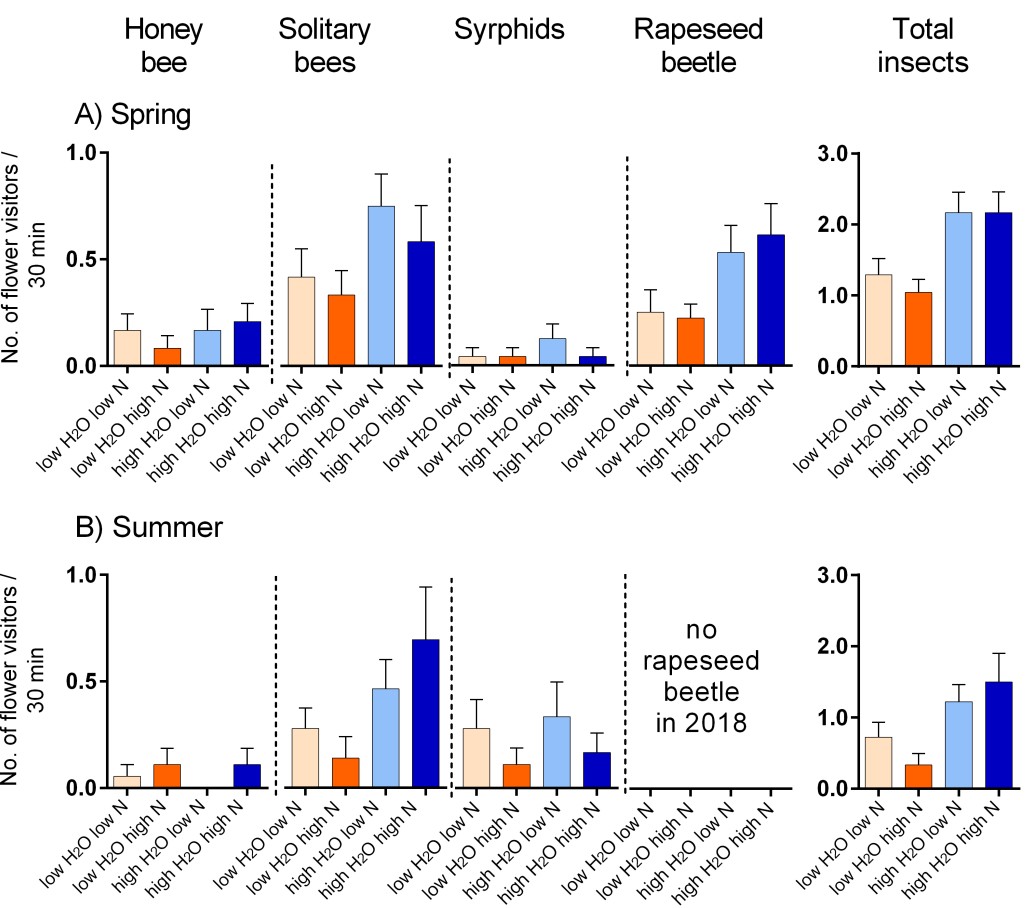

**Figure 5 Flower visitation by major flower visitor groups.** (A) Plants grown in the spring 2017, and (B) in summer 2018. The number of visitors per 30 min. is shown (mean ± SE).

the spring 2017. It also seemed that plants grown in spring with high amount of nitrogen produced a lower number of seeds per flower, while the opposite pattern was apparent in the summer (Fig. 6C), but the interaction of the nitrogen availability and season was not statistically significant ($F = 1.97$, $P = 0.166$). As we showed above, plants grown under different combinations of water and nitrogen availability varied in their total production of flowers. Combined with the variation in the number of seeds produced per flower, this led to differences in the total seed set per plant (Fig. 6D). Specifically, total seed set was higher in plants grown under high water availability, but the effect was stronger in the summer than in the spring (the interaction between water availability and season: $F = 5.03$, $P = 0.029$).

## DISCUSSION

### The effect of environmental changes on plant traits

Our results highlight that water stress is a key factor for both vegetative and floral traits (*Descamps et al., 2018*), while changes of nitrogen supply had a more limited impact in our

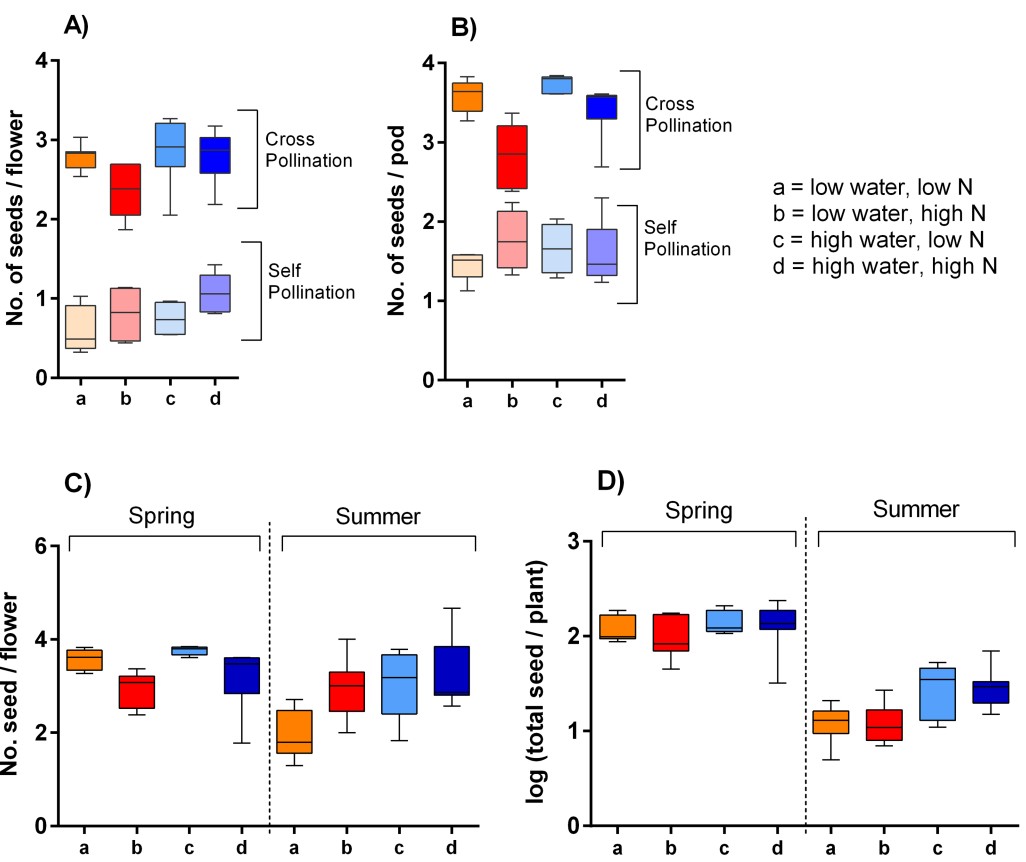

**Figure 6** **Seed production of *S. alba* grown under different growing conditions.** Seed production in plants subjected to self-pollination and cross-pollination by hand: (A) seed production per flower and (B) seed production per pod (median and interquartile range is shown). Seed production of plants subjected to natural pollination in the spring 2017 and the summer 2018: (C) the number of seeds per flower, and (D) the total seed set per plant (median and interquartile range is shown).

case, possibly because the soil mixture we used had a relatively high nitrogen content and water was consequently a more strongly limiting factor. The effects of water and nitrogen availability were mostly interactive and differed between the three time periods (spring 2017, winter 2017-2018, summer 2018). Higher amount of water positively affected plant growth, especially in the spring 2017 and summer 2018. Nitrogen enrichment played a more complex role in the vegetative growth of *S. alba* and its effect was modulated by water availability and differed between the three time periods. Interestingly, previous research has shown that while water deficiency may lead to reduced biomass production and diminished nitrogen uptake in plants (*Cossani, Slafer & Savin, 2012*) and increasing nitrogen supply may enhance their drought tolerance and increase water use efficiency depending on the crop water demand and the irrigation methods (*Quemada & Gabriel, 2016*).

Flowering phenology, the number of flowers, and nectar production of *S. alba* were also affected by growing conditions. In particular, nectar production per flower was affected by a complex interaction of all three variables, where nectar production increased under high

nitrogen availability when water availability was low in the spring 2017 and summer 2018, but not in other cases. Such a complex relationship was not reported by previous studies in other plant species. Several studies showed that nectar production may decline in response to water reduction and increased temperature (*Keasar, Sadeh & Shmida, 2008*; *Scaven & Rafferty, 2013*; *Takkis et al., 2015*). In our case, nectar production did not decrease under these conditions and also showed an opposite result compared to *Hoover et al. (2012)*, where nectar production of *Cucurbita maxima* decreased with higher nitrogen supply and increased with increasing temperature. The comparison of our results with previous studies thus confirms that the effects of varying environmental conditions on nectar production are highly species-specific (*Villarreal & Freeman, 1990*; *Lu et al., 2015*).

Both vegetative and floral traits displayed significant differences among plants grown under the same water and nitrogen supply levels in the three time periods. In particular, plants grown in the winter were smaller and their onset of flowering was significantly delayed compared to the plants grown in the spring and summer. However, we cannot distinguish whether the differences were caused by different temperature, day length, or other factors. Generally, the growth rate and reproductive success of plants is the highest within a certain range of optimal temperatures and decreases rapidly beyond this optimal range (*Vasseur et al., 2014*; *Hatfield & Prueger, 2015*). Phenological shifts in many plants are also closely related to temperature (*Jagadish et al., 2016*; *Kehrberger & Holzschuh, 2019*). Previous studies on *Borago officinalis* also showed that increasing temperature may diminish flower production or lead to flower bud abortion, which may reduce the total number of flowers produced during the plant's flowering period (*Saavedra et al., 2003*; *Descamps et al., 2018*). Similarly, the total number of flowers of *S. alba* significantly dropped in our experiment in the summer 2018 when the temperature in the greenhouse averaged ca. 29 °C. However, the fact that plants produced more flowers with high amount of water, even during the summer, shows that the impact of thermal stress on flower production can be reduced by water supplementation (see also *Mahan, McMicheal & Wanjura, 1995*; *Li et al., 2020*). An optimal temperature is also required for the maximum nectar secretion (*Pacini & Nepi, 2007*; *Lu et al., 2015*). In our case, *S. alba* produced comparatively higher amount of nectar in the spring when the temperature reached intermediate values (average ca. 25 °C). Apart from the temperature, differences in vegetative and floral traits of our plants in the three time periods could have been driven also by differences in day length and light intensity.

## Impact on pollination and seed production

Our experiments showed that differences in traits among plants grown under different environmental conditions had a cascading effect on the number and identity of flower visitors of *S. alba* and on its reproduction. Flower visitation rate of insect-pollinated plants depends on visual cues indicating high floral reward such as the number of open flowers (*Conner & Rush, 1996*; *Akter, Biella & Klecka, 2017*) and the size of floral display (*Grindeland, Sletvold & Ims, 2005*; *Parachnowitsch & Kessler, 2010*; *Biella et al., 2019*), and on the amount and quality of nectar and pollen (*Cresswell, 1999*; *Grindeland, Sletvold & Ims, 2005*). Other morphological features can also influence plant detection by potential

pollinators, such as plant height (*Junker et al., 2013*; *Klecka, Hadrava & Koloušková, 2018a*; *Hernández-Villa et al., 2020*), local plant clustering (*Elliott & Irwin, 2009*; *Akter, Biella & Klecka, 2017*), and flower colour (*Reverté et al., 2016*). Measurements of flower visitation with plants grown in the spring 2017 and plants grown in the summer 2018 revealed that in both cases plants grown with higher amount of water had a significantly higher number of flower visitors compared to plants grown under low amount of water irrespective of nitrogen supply. This is likely a consequence of differences in vegetative and floral traits induced by differences in water availability. As discussed above, plants grown with high amount of water were taller and produced more flowers and these characteristics had a positive effect on the visitation of individual plants as reported in other plant species (*e.g.*, *Mitchell et al., 2004*; *Akter, Biella & Klecka, 2017*; *Klecka, Hadrava & Koloušková, 2018a*). However, other modifications of plant traits induced by water stress also apparently decreased the visitation of plants grown with low amount of water, because the effect of water availability on the number of flower visitors per plant persisted even after accounting for differences in the number of open flowers and plant height in our analysis. We believe that the remaining unexplained variation could be related to nectar chemistry (*Petanidou et al., 2006*; *Hoover et al., 2012*) or flower scent (*Farré-Armengol et al., 2020*).

Besides having lower flower visitation, plants grown under low amount of water had different relative abundance of the main flower visitor groups compared to plants grown with higher amount of water. In both years, the number of solitary bees, hoverflies, other flies, and beetles almost doubled for the plants with high amount of water. In the spring 2017, plants grown with high amount of water received more frequent visits from rapeseed beetles, solitary bees, and hoverflies than plants grown under low amount of water, while the other flower visitors, including honeybees and bumblebees, did not discriminate among the plants. The results were similar in the summer 2018, although rapeseed beetles were almost absent. Although the number of flower visitors differed between the two years the visitation patterns were similar for the plants grown under different conditions. Plants which received higher amount of water were taller and produced higher number of flowers. As a result, they received more flower visitors than plants grown with lower amount of water, except for honeybees in the summer 2018. However, the observed differences in the flower visitation between the spring 2017 and the summer 2018 may be influenced by the differences in overall insect abundance or weather, but not necessarily by the growing conditions of the plants. For instance, increasing temperature may affect flower visitation by a number of mechanisms, from differences in plant traits caused by high temperature stress (*Descamps et al., 2018*), through phenological shifts of plant flowering and pollinator emergence (*Hegland et al., 2009*; *Bartomeus et al., 2011*), to changes in pollinator foraging activity caused by their responses to temperature (*Corbet et al., 1993*; *Slamova, Klecka & Konvicka, 2011*), and direct and indirect effects of temperature on the fitness and mortality of pollinating insects (*Scaven & Rafferty, 2013*). It is also important to note that we are comparing flower visitation between the spring 2017 and summer 2018, so the differences between these sampling periods (*e.g.*, the absence of rapeseed beetles in the summer 2018) could be caused either by seasonality or by inter-annual variation as (*Gómez et al., 2020*) reported that changes in temperature and photoperiod can alter the floral size and shape

and receive a complete different group of pollinators. However, this has no effect on our conclusions about the effects of the water availability and nitrogen supply, which were similar in both periods.

Finally, seed production of *S. alba* was also affected by water and nitrogen availability, apparently both directly through physiological mechanisms and indirectly through changes in insect pollination. Our hand pollination assessment confirmed that *S. alba* is a partially self-incompatible plant (*Fan et al., 2007*). Low water availability reduced seed production per flower in both self-pollinated and cross-pollinated plants, which is consistent with previous studies suggesting that water stress may lead to seed or pod abortion (*e.g.*, *New, Duthion & Turc, 1994*; *Behboudian et al., 2001*). However, we also observed an intriguing effect of nitrogen availability on seed set: increased nitrogen availability increased seed set in self-pollinated plants, but decreased seed set in cross-pollinated plants. We are not aware of any studies which would show that high nitrogen supply can cause seed abortion.

Seed count per flower from the naturally pollinated plants in the spring 2017 also showed a similar trend as in plants cross-pollinated by hand, where the number of seeds per flower increased in plants grown with high water availability but decreased with high nitrogen availability. In contrast, in experiments done in the summer 2018, the number of seeds per flower was not affected by nitrogen availability and decreased in plants grown with high amount of water. Total seed set per plant was unaffected by nitrogen availability and increased in plants grown under high water availability –moderately in the spring 2017 but much more in the summer 2018. This may stem from differences in the composition of the flower visitor community between plants grown under low and high water availability and from the higher total visitation rate in the spring 2017 compared to summer 2018. The level of pollen limitation (*Knight et al., 2005*) was thus higher in the summer 2018, which likely explains why the number of seeds per flower was lower and was more strongly reduced in plants grown with low amount of water.

## CONCLUSIONS

We have shown that multiple environmental factors have a complex and interactive impact on plant traits, visitation by pollinators, and seed production. Our model species, *S. alba*, is an important crop and a close relative to many other economically important crops and vegetables from the Brassicaceae family, hence our experiment shows how different climatic factors may affect both vegetative growth and crop yield in plants form this family in the future extreme climatic events. We conclude that not only increasing temperature, but also reduced precipitation and nitrogen enrichment, may impact plant–pollinator interactions with negative consequences for the reproduction of wild plants as well as insect-pollinated crops.

## ACKNOWLEDGEMENTS

We are grateful to Pavla Koloušková for her assistance with the field observations and Jana Jersáková for her help with the laboratory analyses.

### Funding

This study was supported by the Czech Science Foundation (Grant no. GJ17-24795Y). The funders had no role in study design, data collection and analysis, decision to publish, or preparation of the manuscript.

### Grant Disclosures

The following grant information was disclosed by the authors:
The Czech Science Foundation: GJ17-24795Y.

### Competing Interests

The authors declare there are no competing interests.

### Author Contributions

- Asma Akter conceived and designed the experiments, performed the experiments, analyzed the data, prepared figures and/or tables, authored or reviewed drafts of the paper, and approved the final draft.
- Jan Klečka conceived and designed the experiments, analyzed the data, authored or reviewed drafts of the paper, and approved the final draft.

### Data Availability

The data is available at figshare: Akter, Asma; Klečka, Jan (2020): Interactive effects of temperature, water, and nitrogen availability on the growth, floral traits, and pollination of white mustard, Sinapis alba. figshare. Dataset. https://doi.org/10.6084/m9.figshare.13317686.v1.

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
