# Peer review of "Water stress and nitrogen supply affect floral traits and pollination of the white mustard, Sinapis alba (Brassicaceae)"

_PeerJ, doi:10.7717/peerj.13009_

## Round 0.1 · original submission · Major Revisions

The manuscript addresses a very interesting subject, is well presented and well written. However, according to the reviewers, I must recommend doing some revision before publication.

I acknowledge the merits of the manuscript highlighted by the first reviewer. My main concern is that the second level of applied nitrogen is too high and no data on the available nitrogen in the growing medium are presented. It seems that the first nitrogen level is close to optimum. That is, in my opinion, why no positive responses to increasing N level were observed. It is unfortunate that a lower nitrogen rate has not been tested. However, in the context of global change, increases in nitrogen inputs to soil could occur. In that case, according to your results, no beneficial effects of those increases would be observed and even adverse effects could occur. This fact should be emphasized to justify the nitrogen levels tested.

I rather agree with reviewer 1 that the Introduction is strong and organised. However, it is up to you to consider shortening it as suggested by reviewer 2.

Regarding the use of a different statistical modelling, suggested by reviewer 2, you must assess whether this would result in an improvement of the information obtained. However, I think that the season is an important factor, since it allows to assess the effect of temperature, even if it is from different years.

I include an annotated manuscript with my suggestions.

·

Basic reporting

In my basic reports I, Helena Maura Torezan-Silingardi, want to congratulate the authors that chose to investigate a super relevant theme, which is up to date mainly considering the recent and gradual climate modifications we observe. The manuscript points direct and indirect consequences on plant species and their associated flower visitors and fruit production. The results obtained here to a cultivated species offer us consistent information. I think this manuscript soon will be an article cited by many other researchers, including myself. It was a pleasure to read this manuscript.

Now I will present my comments about each part of the manuscript. The language is unambiguous and professional, the literature is adequate. The abstract is concise and points the most relevant information, but the keywords are missing. The introduction is very strong and organized, with recent, important literature and clear and well-defined objectives. The experimental design is interesting, even so I suggest improving it adding some few information in the methodology, as I point in detail below. The results are described with high quality figures and tables, all of them very informative and clear and well explained in the text, so I consider the validity of the findings are very good and clear. The discussion is extensive and contain relevant comparisons with many other studies. The conclusions are concise and valid.

General comments: I indicate minor revision as the manuscript is relevant, well planned and executed.
Confidential notes to the editor: The manuscript is relevant, very good, and will be cited by many researchers. The modifications I ask are few and simple to be done.

Here I present my suggestions for each line numbered:
Line 32 – Authors wrote the “nitrogen availability played an important role in nectar production”, what is presented also in lines … and in Figure 2c. This information seems to be imprecise, as Figure 2c presents a higher nectar volume restricted to just one treatment and one season: low water/high N in Spring, but neither in any other treatment nor in Winter or Summer time. So I suggest the authors to make the text more precise, defining the exact treatment and the season which presented the distinct benefits of N supplementation.
Line 122 – If the main insects visiting the flowers are Apis mellifera, bumble bees and solitary bees in Europe, were the observations were made, I expected to have these same kind of visitors observed. But bumble bees were absent. Could you explain?
Lines 127-128 – Please, explain the reason you used a distinct number of plants (30, 60 and 45) in each of the seasons investigated. Are the distinct numbers related to diseases and aphids, as mentioned in line 150 ?
Lines 134-135 –I think all the seeds used in the same season received water in the germination trays in same day. Is this supposition correct? Could you detail a little bit more the germination process, please?
Lines 144-145 – I suppose the N quantities should be described in the same way, and not as “0,242 g N pot-1 (~100kg N ha-1)” and “0,121 g N pot-1 (~50 kg N pot-1)”. Please, check them.
Line 153 – I think the authors intended to write ‘floral traits’, but accidently they wrote “flora traits”.
Lines 155-156 – Could you explain your decision about counting the number of leaves only in the main shoot and not in the entire plant, please? Is it possible that all the distinct treatments had plants with the same number of shoots? The steam diameter was measured in just one shoot, I suppose the higher one, is it correct? This information could be detailed better.
Line 171-172 – Please, indicate the number of flowers used in each treatment.
Line 173 – Please, indicate how many days you had to wait from pollination to the completion of seed development.
Line 223 – Observing Figure 2A it seems wrong to say that the onset of flowering was “slightly delayed also by high nitrogen supply”. I suppose there is not a significative statistical difference on the flowering time.
Line 225 – The number of flowers produced in winter does not seems to be “slightly negatively by nitrogen supply” with high water.
Line 273-274 – Please, inform that in Spring the plants grown under higher nitrogen availability produced a lower number of seeds per flower, but the results were not the same in Summer.
Lines 294-296 – Nectar production was significatively increased in Spring with higher nitrogen and low water supply. But not in any other treatment.
Line 339 – Please, consider removing the ‘and’ from “plants grown in the summer 2018 revealed that and in both cases…”.
Lines 352-353 – There was not a different composition of flower visitor conditioned by water supply.
* * *
Experimental design

The experimental design is interesting and extensive, performed with rigourous care. Even so I suggest improving it adding some few information in the methodology to make possible to replicate it perfectly, as I pointed in detail in the basic reporting.
The research question is relevant, clear and meaningful. The results are properly presented with great figures and the discussion is extensive.
The manuscript do not fail to meet the PeerJ standards.

Validity of the findings

The findings are clearly presented and discussed, they are linked to each of the objectives proposed. The discussion presents many studies properly related to the present one. After some minor ajustements in the text the experiment can be repeated by any other reseacher in the word.

Additional comments

I suggested minor modifications in the manuscript to improve it's quality, They will be done easily by you. The manuscript is very interesting and will be readed by many of researchers, including in my laboratory.

Reviewer 2 ·

Basic reporting

English needs improvement
Some important references are missing (v.g. Majetic 2017 Flora, Rushman et al. 2019 Plant Cell Environ, Gomez et al. 2020 Nat Comm.)

Experimental design

It seems that experiments were performed in different years: spring 2017, winter 2017-2018 and summer 2018. This is unfortunate, since it means that authors may be confounding years with season. This is specially true for that part of the study performed outside the greenhouse. So, for example, it is hard to tell how much of the seasonal variation in visitor abundance and identity is due to season itself or, on the contrary, is due to yearly variation (what it is well known to occur in many other systems, including pollinator-generalized crucifers). I am afraid that this fact precludes any possibility of considering season a main factor in the experiment.
Overall, statistical analyses are fine. However, the experimental design suggests that a different staristical modelling should have done. So, season was not manipulated, and if I am right plants were all of them located in the same greenhouse. If this how the experiment was done (it is not clear from the description if there were on or several greenhouses), I think that the rightful way of analyzing the data is by means of a split-plot model (or a repeated-measure model, they are analytically similar), where greenhouse is the main plot and the two manipulated factors are nested into it. Consequently, no triple interaction is possible, and no conclusion about the effect of season can be reached.
Putting together these two concerns, authors should consider removing from their narrative any mention to season. The design of the experiment impedes any possibility of considering season as a factor at the same explanatory level as water and nitrogen.

Validity of the findings

There were different number of experimental plants in each season.mThis may also have consequences for how data should be analyzed.

Additional comments

Review of manuscript entitled ‘water stress and nitrogen supply affect floral traits and pollination in the white mustard, Sinapis alba (Brassicacea).

In this experimental work, authors checked the effect of altering water and nitrogen on some plant traits and the interaction with pollinators. Results were interesting, since they found that these two factors impacted flower visitation rate and affected seed production. Overall, the study is well conducted and analyzed, and the conclusions are interesting, highlighting the importance that the abiotic context has on the pollination process in many plants. I have however some concerns on both the methodology and the analysis.
Main concerns
It seems that experiments were performed in different years: spring 2017, winter 2017-2018 and summer 2018. This is unfortunate, since it means that authors may be confounding years with season. This is specially true for that part of the study performed outside the greenhouse. So, for example, it is hard to tell how much of the seasonal variation in visitor abundance and identity is due to season itself or, on the contrary, is due to yearly variation (what it is well known to occur in many other systems, including pollinator-generalized crucifers). I am afraid that this fact precludes any possibility of considering season a main factor in the experiment.

Overall, statistical analyses are fine. However, the experimental design suggests that a different staristical modelling should have done. So, season was not manipulated, and if I am right plants were all of them located in the same greenhouse. If this how the experiment was done (it is not clear from the description if there were on or several greenhouses), I think that the rightful way of analyzing the data is by means of a split-plot model (or a repeated-measure model, they are analytically similar), where greenhouse is the main plot and the two manipulated factors are nested into it. Consequently, no triple interaction is possible, and no conclusion about the effect of season can be reached.

Putting together these two concerns, authors should consider removing from their narrative any mention to season. The design of the experiment impedes any possibility of considering season as a factor at the same explanatory level as water and nitrogen.

Other concerns
Introduction can be shortened. For example, the sentences “For example, several studies on fruit and vegetable plant species showed that pollinator preferences for their flowers strongly depended on nectar volume or sugar concentration (Jabłoński et al., 1984; Schmidt et al., 2015; Roldán-Serrano and Guerra-Sanz, 2005). A study on Citrus plants also showed that honeybee visitation rate was positively correlated with nectar volume (Albrigo et al., 2012). These investigations indicated that both the quantity and quality of nectar influences flower preferences of pollinators” can be deleted.

Why were there different number of experimental plants in each season? This may also have consequences for how data should be analyzed.

Authors should consider mentioning some recent studies published on floral plasticity and its consequences on floral visitors, such as Majetic 2017 Flora, Rushman et al. 2019 Plant Cell Environ, Gomez et al. 2020 Nat Comm.

---

## Round 0.2 · Major Revisions

You have answered to the reviewers' comments but have not addressed ANY of my own comments. So I have to recommend major revisions again. Please read my annotations to v0 (I enclose again an annotated manuscript of v1) and address all my suggestions.
Moreover, take into account the concerns that reviewer 2 still have regarding the statistical model.

Reviewer 2 ·

Basic reporting

I want to thank the authors for their thorough review and for having added all our suggestions. The ms is much improved now. I have nothing to say on this point.

Experimental design

I think authors have done a good job in discussing their results in this new version. However, i still think authors have not analyzed the experiment using the most appropriate statistical model. I am sorry for previous misunderstanding. I did not mean using a repeated-measure model, as authors seemed to understand (if you go to my first review, you can see clearly that i included rm-model within parenthesis, just to indicate that they are analytically similar to other more appropriate statistical models). I still think that authors could do a better job using any type of model that assume some kind of constrain in the random assignment of experimental units, rather than the fully-factorial model they have used (with all the consequences this have for the interpretation of the results, that although two different issues they are tightly interlinked). I suggested split-plot analyses in my previous review, no rm analyses, but other models including nested variables could be appropriate too. Nevertheless, if authors consider 3-way GLM is appropriate to analyze their experiment, it is OK with me.

Validity of the findings

I think the findings are very interesting, and well discussed. I do not have anything else to add.

---

## Round 0.3 · accepted · Accept

The manuscript has considerabli improved, being acceptable for publication.

Please, during proofreading, in lines 134-135 write Nitrogen and Phosphorus in lower case (nitrogen and phosphorus) and specify if the concentrations are available or total nitrogen and phosphorus (I suppose they are total).